# Scale and space dependencies of soil nitrogen variability

Ana M. Tarquis[1,2], María Teresa Castellanos[3], Maria Carmen Cartagena[3], Augusto Arce[3], Francisco. Ribas[4], María Jesús Cabello[4], Juan López de Herrera[1] and Nigel R.A. Bird[1]

[1]CEIGRAM, Ciudad Universitaria sn, Madrid, 28040, Spain.- Technical University of Madrid.

[2]Dpto. Matemática Aplicada - E.T.S.I. A.A.B. -Technical University of Madrid, Spain.

[3]Dpto. Química y Tecnología de Alimentos- E.T.S.I. A.A.B - Technical University of

10    Madrid, Spain.

[4]Centro de Investigación Agroambiental El Chaparrillo – Inst. Regional de Investigación y Desarrollo Agroforestal (IRIAF), Ciudad Real, Spain.

*Correspondence to:* Ana M. Tarquis (anamaria.tarquis@upm.es)

**Abstract:** In this study, we use multifractal analysis, through generalized dimensions ($D_q$) and the relative entropy ($E(\delta)$), to investigate the residual effects of fertigation treatments applied to a previous crop on wheat and grain biomass and nitrogen content. The wheat crop covered nine subplots from a previous experiment on melon responses

20    to fertigation. Each subplot had previously received a different level of applied nitrogen (Napp), and the plants from the previous melon crop had already taken up part of it. Many factors affect these variables, causing them to vary at different scales and creating a non-uniform distribution along a transect. Correlations between the four variables and Napp showed high volatility, although the relationships between grain weight and wheat

25    weight versus wheat nitrogen content presented a statistically significant logarithmic trend.

The $D_q$ values were used to study the relation between scales, and $E(\delta)$ values and their increments between scales were used to identify the scale at which the variable had the maximum structure and were compared with the scaling behaviour of the Napp. $E(\delta)$ is particularly appropriate for this purpose because it does not require any prior assumptions regarding the structure of the data and is easy to calculate.

The four variables studied presented a weak multifractal character with a low variation in $D_q$ values, although there was a distinction between variables related to nitrogen content and weight. On the other hand, the $E(\delta)$ and the increments in $E(\delta)$ help us to detect changes in the scaling behaviour of all the variables studied. In this respect, the results showed that the Napp through fertigation dominated the wheat and grain biomass response, as well as the nitrogen content of the whole plant; surprisingly, the grain nitrogen content did not show the same structure as Napp. At the same time, there was a noticeable structure variation in all the variables, except wheat nitrogen content, at smaller scales that could correspond to the previous cropping root arrangement due to uptake of the Napp.

**Key words**: relative entropy, multifractal analysis, sink crop

## 1. Introduction

Soils exhibit spatial variation operating over several scales. This observation points to "variability" as a key soil attribute that should be studied (Burrough et al., 1994). Soil variability has often been considered to consist of "functional" (explained) variations plus random fluctuations or noise (Goovaerts, 1997 and 1998). However, the distinction between these two components is scale-dependent because increasing the scale of observation almost always reveals structure in the noise (Logsdon et al., 2008). Geostatistical methods and, more recently, multifractal/wavelet techniques have been used to characterize the scaling and heterogeneity of soil properties, among other approaches coming from complexity science (de Bartolo et al., 2011). These methods study the structure of the property measured in the sense that compares the probability distribution at each scale and among scales.

Multifractal formalism, first proposed by Mandelbrot (1982), is suitable for variables with self-similar distribution on a spatial domain (Kravchenko et al., 2002). Multifractal analysis can provide insight into spatial variability of crop or soil parameters (Kravchenko et al., 2002 and 2003; Vereecken et al., 2007). This technique has been used to characterize the scaling properties of a variable measured along a transect as a mass distribution of a statistical measure on a spatial domain of the studied field (Zeleke and Si, 2004; López de Herrera, 2016). To do this, it divides the transect into a number of self-similar segments. It identifies the differences among the subsets by using a wide range of statistical moments.

Wavelets were developed in the 1980s for signal processing, and later introduced to soil science by Lark and Webster (1999). The wavelet transform decomposes a series; whether this be a time series (Whitcher, 1998; Percival and

Walden, 2000), or as in our case a series of measurements made along a transect; into components (wavelet coefficients) which describe local variation in the series at different scale (or frequency) intervals, giving up only some resolution in space (Lark et al., 2003). Wavelet coefficients can be used to estimate scale specific components of variation and correlation. This allows us to see which scales contribute most to signal variation, or to see at which scales signals are most correlated (Lark et al, 2004). This can give us an insight into the dominant processes.

An alternative to both of the above methods has been described recently. Relative entropy and increments in relative entropy has been applied in soil images (Bird et al., 2006) and in soil transect data (Tarquis et al., 2008) to study scale effects localized in scale and provide the information that is complementary to the information about scale dependencies found across a range of scales. We will use them in this work to describe the spatial scaling properties of a set of data measured on a common 80-m transect across a wheat crop field. This is an indirect way to study the N variability left in the soil by the previous crop.

Nitrogen fertilizer inputs for intensive production of irrigated crops can contribute to elevated $NO_3$ concentrations in groundwater when crop N use is insufficient to deplete the available soil N. The practice of drip fertigation has the potential to increase the efficiency of water and nitrogen use efficiency (Castellanos et al., 2010). However, a disadvantage associated with it is that the nitrogen travels outside the root zone (Thompson and Doerge, 1996). Other workers have investigated the residual effects of nitrogen (McCracken et al., 1989; Karlen et al., 1998; Ruffo et al., 2004; Bundy and Andraski, 2005). The accumulation and redistribution of nitrogen within the soil varies depending on management practices, soil characteristics and

precipitation, and these effects are likely to contribute to variations at different spatial frequencies. None of the studies of which we are aware consider the effects of previous treatments over a range of spatial frequencies, and given the particular processes associated with fertigation, we wished to do so in this study.

The data discussed in this paper result from two consecutive experiments performed near two hydrological units (UH) protected by the government of Castilla-La Mancha concerning the protection of waters against pollution caused by nitrates from agricultural sources. These two units, Mancha Occidental (UH.04.04, 6.953 km2) and Campo de Montiel (U.H. 04.06, 3,192 km$^2$), have been declared vulnerable zones to
nitrate pollution with high $NO_3$ contamination problems. In the first experiment, the plots were used for melon crop experiments to optimize fertigation using different levels of N, as reported in Castellanos et al. (2010). These treatments constituted a known contribution to the variation of soil nitrogen at predominantly larger scales. During melon crop development, a proportion of the nitrogen was taken up, adding a
second factor of variability that is also known at smaller scales. After the melons were harvested, the second experiment with wheat was begun. Wheat was sown across the plots and harvested in consecutive sections along the transect and biomass, and the N uptake was measured. The wheat was used effectively as a nitrogen sink crop and allowed us to evaluate the residual soil nitrogen.

In this study, we have analysed the transect data for nitrogen content and the weight of the grain and of the whole plant of the wheat crop. First, correlations between these four variables and the different nitrogen application doses in the previous crop were estimated, without considering spatial structure. Then, multifractal and relative

entropy analyses were applied to investigate the structure among the scales. This work is the first application of both types of analysis to the same data set.

## 2. Materials and Methods

### 2.1. Field Experiment

Field trials were conducted in *La Entresierra* field station of Ciudad Real in the central region of Spain (3° 56' W; 39° 0' N; 640 m of altitude) during May 2006 to June 2007. The soil of the experimental site, classified as Petrocalcic Palexeralfs in the USDA system (Soil Survey Staff, 2010), presented very low vertical variability up to a depth of 60 cm, from which one finds a discontinuous and fragmented petrocalcic horizon. The soil was sandy-loam in texture, moderately basic (pH 7.9), with a medium level of organic matter (2.2%), rich in potassium (0.9-1.0 meq $L^{-1}$, ammonium acetate) and with a medium level of phosphorous (16.4 to 19.4 ppm, Olsen) with ECw. 0.1-0.2 $dS\ m^{-1}$.

The area is characterized by a continental Mediterranean climate, with widely fluctuating daily temperatures (for more details, see Castellanos et al., 2010).

During the three years prior to this experiment, the plots did not receive any organic or fertilizer amendments and were used to grow non-irrigated winter wheat (*Triticum aestivum* L.).

### 2.2. Melon Crop Experiment

In this experiment, a randomized complete block design was used, with three nitrogen treatments and three irrigations. The irrigation treatment was applied at the

main plot level, and N-rates were replicated in the subplots. Each treatment was replicated four times in subplots measuring between 7.5-16.5 m in width and 12 m in length. The subplot widths ranged in size for practical reasons. The plots were arranged on a four by nine grid (Fig. 1). Each subplot had five, seven or eleven rows of melons, according to its width (see Fig. 1).

Each crop row was drip irrigated from a line with emitters spaced at 0.5 m, which dripped water at a rate of 2 l h$^{-1}$. Initially, to facilitate crop establishment, all plots received 30 mm of water. The irrigation schedule was calculated from June 8 to September 6, with a single daily irrigation of 60% (W1), 100% (W2) or 140% (W3) of melon crop evapotranspiration (ETc) depending on the irrigation treatment. Crop evapotranspiration (ETc) was calculated daily following the FAO method (Doorenbos and Pruitt 1977) as follows:

$$ETc = Kc \times ETo \qquad (1)$$

where Kc is the crop coefficient, which was obtained in the same area for the melon crop in earlier years (Ribas et al. 1995), and ETo is the reference evapotranspiration calculated by the FAO Penman-Monteith method (Allen et al. 2002) using daily data from a meteorological station sited near the experimental field. The rainfall was negligible during the crop experiment, so the water applied was calculated as the ratio between the ETc of the previous week and the efficiency of the system, which considers the salt tolerance of the crop, the quality of the irrigation, the soil texture and the homogeneity of the irrigation system (Rincón and Giménez, 1989), estimated as 0,81 under the study conditions (more details in Castellanos et al., 2010). The irrigation calculated in this form was the theoretical irrigation and was divided by the number of days to obtain daily irrigation requirements. The total irrigation applied

was registered on the water meter. The ETo during the irrigation schedule was 572 mm, the ETc was 419 mm, and the total irrigation applied was 343, 553 and 756 mm for W1, W2 and W3, respectively (Table 1). The irrigation water quality was measured weakly through a chemical analysis to estimate the nitrogen content of the water (*Nw*) (Table 1).

The fertilizer treatments consisted of different N doses: 0 (N0), 150 (N1) and 300 (N2) kg ha$^{-1}$. The N fertilizer was applied in the form of ammonium nitrate from June 9 to August 18, from a single pool at one end of the field where irrigation water was mixed with the respective doses of N (Table 1). The total amount of N applied was the sum of the N fertilizer and N in the irrigation water, so all the treatments appear in Table 1.

The plots were fertilized with 120 kg of $P_2O_5$ ha$^{-1}$ (phosphoric acid) for the season, added to the irrigation water and injected daily from June 8 to August 30.

Melons were harvested when there was a significant amount of ripe fruit in the field from 26 July to 7 September, with a total of seven harvests.

The duration of the melon experiment was from May 24 to September 7, and it is described more fully in Castellanos et al. (2010).

**2.3. Wheat Crop Experiment**

Winter wheat (cv. Soissons) was grown on the same experimental sites where the melon crop was before (Fig. 2). It was sown 20 December of 2006 in rows spaced 0.15 m apart at a population of 400 seeds m$^{-2}$. Post emergence herbicides were used to

control weeds. No fertilizer or organic amendments were use for the cereal crop. Wheat crop was harvested 6 June 2007.

At this time a transect was selected in the field that went through several plot treatments as showed in Fig.1. Each 0.5 m a frame of 0.5 x 0.5 m$^2$ was placed on the soil and the wheat plants captured were harvested and placed in labelled samples. A total of 160 samples were collected traversing a length of 80 m.

Sub-samples of the dry plants and wheat grain were ground to a fine powder to determine the N content using the Kjeldahl method (Association of Official Analytical Chemists, 1990).The N uptake by the plant (*PN*) and by the grain (*GN*) was obtained as a product between N concentration and biomass (*PW* and G*W*, respectively). The resulted data is showed 5 in Fig. 3A and 3C.

In each sample, the wheat grain was placed apart from the rest of the plant to obtain the dry weight of each sample separately. The grain dry weight (*GW)* and plant dry biomass were determined by oven drying at 80 °C to constant weight. The plant dry weight (*PW*) was the sum of the *GW* and plant biomass. The data are shown in Fig. 3B and 3D.

## 2.4. Correlations

A simple analysis, regardless of spatial position, were applied to the data collected. The correlation (r) and the determination coefficient ($R^2$) between the nitrogen applied during the melon crop (Napp) and each variable (*PW*, *GW*, *PN* and *GN*) were estimated and plotted.

At the same time, the relations between nitrogen content and weight were studied for the grain (*GW* versus *GN*) and the whole plant (*PW* versus *PN*) as well as *GW* versus *PN* to compare with other studies performed in wheat crops.

Finally, a statistical test was applied for each variable to determine if there was any significant trend with distance that would not allow the application of a straight multifractal analysis on the original data. The measure used was the coefficient of the slope of the regression line along the distance. This coefficient is derived using the least squares method and then compared to zero using the Student t-test. If the t value is less than a critical t value at the 95% level for the degrees of freedom, then the slope is considered to be zero.

## 2.5. Multiscale analysis through Generalized Dimensions

The aim of a multifractal analysis (MFA) is to study how a normalized probability distribution of a variable ($\mu_i$) varies with scale as it is one way to study the structure of a measure. In this sense, the density levels of these probabilities are evaluated through the behaviour of a range of statistical moments of the partition function ($\chi(q,\delta)$). Let's consider a grid segment of length $\delta$ covering a part of transect, with total length $L$. The measure of the i$^{th}$ segment is defined $M_i(\delta)$. The probability is:

$$\mu_i(q,\delta) = \frac{M_i^q(\delta)}{\sum_{j=1}^{N(\delta)} M_j^q(\delta)} \tag{2}$$

For a multifractal measure, $\chi(q,\delta)$ will have scaling properties (Evertsz and Mandelbrot, 1992), namely

$$\chi(q,\delta) \sim \delta^{\tau(q)} \qquad (3)$$

$$\text{Being } \chi(q,\delta) = \sum_{j=1}^{N(\delta)} \mu_j^q (\delta) \qquad (4)$$

where $\tau(q)$ is a nonlinear function of $q$ (Feder, 1989). For each $q$, $\tau(q)$ may be obtained as the slope of a log-log plot of $\chi(q,\delta)$ against $\delta$. A generalized dimension function $D_q$ is then derived as (Hentschel and Procaccia, 1983):

$$D_q = \tau(q)/(1-q) \qquad (5)$$

for $q \neq 1$. The case $D_1$ is defined as the limit $D_1 = \lim_{q \to 1} D_q$. This leads to the scaling relation of entropy given by:

$$S(\delta) = -\sum_{i=1}^{n(\delta)} \mu_i \ln(\mu_i) \sim D_1 \ln(\delta) \qquad (6)$$

The dimension $D_1$, known as entropy dimension, can then be extracted from a plot of entropy against $\ln(\delta)$.

## 2.6. Multiscale analysis through Relative Entropy

Given these definitions and the behaviour to expect in case of a multifractal measure, we are going to focus in the scaling properties of entropy as a tool to quantify the heterogeneity of coarse grained measure $\mu_i(\delta)$, or signal, derived from the transect data as it has been applied previously to black and white soil thin sections (Bird et. al., 2006).

We consider a transect of length $L$ for a bin size $\delta$ the entropy ($S(\delta)$) is defined by equation (6). We use here a relative entropy ($E(\delta)$) in order to establish what difference exists from the entropy of a uniform measure, given by

$$E(\delta) = \sum_i \mu_i(\delta) \ln \mu_i(\delta) - \ln \frac{\delta}{L} \qquad (7)$$

where the second term is the entropy of the uniform measure. Plotting this against the resolution of observation $\delta$, then reveals how heterogeneity in the signal evolves with increasing resolution being another way to study the structure of the measure or variable (Tarquis et al., 2008). We may use this simple procedure to identify multiscale signals arising from the superposition of structure at different scales and

assess the degree of this scale dependent structure.

Here we consider some special cases. When we increase the resolution by a factor of 2 we observe that

$$E(\delta/2) = E(\delta) + \sum_i \mu_i (p_i \ln p_i + q_i \ln q_i) \qquad (8)$$

where $p$ and $q$ control the distribution of the measure in the finer partition and

$p + q = 1$. Then

$$\Delta E(\delta) = E(\delta/2) - E(\delta) = \sum_i \mu_i (p_i \ln p_i + q_i \ln q_i) + \ln 2 \qquad (9)$$

If $p$ and $q$ are independent of $i$ then

$$\Delta E(\delta) = (p_i \ln p_i + q_i \ln q_i) + \ln 2 \qquad (10)$$

This increases as the difference between $p$ and $q$ increases and more structure is

observed in the data at this scale. If $p = q = 0.5$, namely there is no structure revealed on increasing resolution then $\Delta E(\delta) = 0$.

Further if $p$ and $q$ are independent of $\delta$ then we arrive at a binomial cascade. This is a multifractal measure and relative entropy scales logarithmically as:

$$E(\delta) = (D_1 - 1)\ln(\delta / L) \qquad (11)$$

## 3. Results and Discussion

### 3.1. Correlations

Classical statistical analyses were performed on each of the variables to study their first statistical moments (Table 2). We could observe that the average and median present differences for each variable, in contrast to a normal distribution where both coincide. However, kurtosis and asymmetry do not present values higher than the unit in absolute terms. *GW* and *PW* present the highest kurtosis (0.82 and 0.78) and are negative. On the other hand, *GN* and *PN* have the highest asymmetry and are positive. The coefficient of variation is higher in variables related to nitrogen content (*GN* and *PN*) and lower in variables related to weight (*GW* and *PW*).

To study the relationships of *GW*, *PW*, *GN* and *PN* with the nitrogen applied during the melon crop season (Napp), we have plotted these variables without considering any spatial factors (Fig. 4). All of them show a tendency, as we expected, to increase in value as Napp increases. The correlation coefficient (r) for the four variables range from 0.66 (*GN* case) up to 0.77 (*PN* case) demonstrating that there are statistically significant correlations with the N application in the melon crop experiment (Napp), as the wheat crop did not receive any N directly. For this reason, the relationship that we can observe could be considered linear, as the range we are studying is suboptimal and not as in other studies (e.g., Hawkesford, 2014). However, a quadratic relation can be fitted to all the variables with a similar $R^2$ (results not shown).

However, we can observe that at each of the Napp values, the variables show variability. This result is a consequence of a set of processes occurring from melon fertigation to wheat harvest, such as nitrogen uptake by the melon crop, organic soil nitrogen mineralization, nitrogen leaching, horizontal diffusion of soluble nitrogen forms and nitrogen uptake by the wheat crop (Milne et al., 2010).

The positive effect of increasing grain weight together with the additional benefit of increasing wheat N content with increasing N application is shown in Fig. 5A. Moreover, the same positive effect of N addition was observed, increasing wheat weight together with increasing wheat N content (Fig. 5B). Closer inspection of Fig. 4 reveals that the variability was much higher when the N application was higher. Barraclough et al. (2010), in an experiment with N fertilization applied homogenously directly to the wheat crop, found that much of the additional N taken up by the plant (PN) is manifested in higher yield (GW), although we remark again that in this work, the N application was performed in the melon crop experiment, through fertigation on crop lines, and the wheat crop did not receive any N fertilization and was not irrigated.

This positive effect of N addition has been observed in numerous studies (Barraclough et al., 2010 and references therein). Several works determine the N optimum in the wheat crop, but in this study, the optimal N dose was not obtained because we sought to study the variability and the effect of the residual N resulting from N application to a previous melon crop months before.

Before applying the multifractal analysis, a statistical test was applied to each variable to determine whether it presented a significant trend with distance. The results are shown in Table 3, where the estimated t was always lower than the critical t-value, implying that no spatial trend was significant.

### 3.2. Generalized Dimensions

Multifractal analysis was applied to the four variables. In all cases, a $\tau(q)$ function reflected a hierarchical structure from one scale to the other with values of q=1, $\tau(1)=0$, indicating the conservative character of the variables (Fig. 6A). Therefore, we estimated the $D_q$ in an interval of q=±4 (Fig. 6B). The results show a weak variation in the values near 1, highlighting the difficulty of characterizing the multiscale heterogeneity in this type of analysis. In this case, the scale dependency found across a range of scales is not strong enough to show a high variation in $D_q$ versus q, and $\tau(q)$ presents an almost linear trend. There are several works on soil transect data that present similar results (Caniego et al., 2005; Zeleke and Si, 2006).

Calculating the difference of $D_{-4}$ and $D_4$ can provide an estimate of the variation of $D_q$ for each variable. A higher difference implies a stronger multifractal character. The variables related to nitrogen content (*GN* and *PN*) show a higher variation in $D_q$ values (0.151 and 0.150, respectively) than the variables related to weight (0.088 for *GW* and 0.092 for *PW*), highlighting a different multifractal character of the two types of variables. In this sense, *GW* and *PW* behave very similarly, as do *GN* and *PN*. This information is complementary to the descriptive statistics performed in section 3.1, in which the spatial factor was not considered.

### 3.3. Relative Entropy

To compare the spatial scaling behaviour of these four variables with the Napp behaviour, $E(\delta)$ was calculated, and the results are shown in Fig. 7A and Fig. 8. The translation from the number of data points ($\delta$= 1, 2, 5, 10, 20, 40, 80 and 160) to the

distance in metres is marked in Fig 7A. The trend in each case is not log-linear, as we would expect for a pure multifractal measure. In the case of Napp, the range of values reached -0.20 (Fig. 7A), and in the rest, they approach -0.06 (*GW* and *PW*) or -0.11 (*GN* and *PN*) (Fig. 8).

We have plotted each variable (Fig. 8) $E(\delta)$ calculated at each $\delta$ following equation [7] and based on $D_1$ estimated in the above section using equation [11]. At certain scales, both present the same value, but most of the scales show variations (Fig. 8). Comparing the straight line slopes (see Fig. 8), which derived from the $D_1$ values, higher and very similar values are found in GN and PN. On the other hand, GW and PW

present lower values and are very similar to each other.

The increments of the $E(\delta)$ ($\Delta E(\delta)$), between two consecutives scales, calculated for Napp and the four variables are shown in Fig. 7B and Fig. 9, respectively. *PN*, *GW* and *PW* present a similar scaling trend, with a maximum structure revealed at scale $\delta=10$, corresponding to a distance of 5 m. This behaviour is the same found in Napp in

the melon crop. In the case of *GN,* the maximum structure is found at $\delta=20$ (10 m), indicating that the interaction of other factors influences in this variation, and the Napp is not the main one.

All the values of $\Delta E(\delta)$ at the smallest scales, $\delta=5$, 2 and 1 (2.5, 1 and 0.5 m respectively), show an increase, giving the second maximum value for *GN*, *GW* and

*PW.* This result suggests that at those scales, the variation is mainly due to the melon cropping lines, as the uptake of the applied nitrogen by this crop left a lower amount of available nitrogen for the wheat crop. In the case of *PN,* the second maximum was found at $\delta=20$ (10 m) followed by the one at the smallest scales, $\delta=2$ and 1 (1 and 0.5 m), as in the other variables.

Comparing these results with those published by Milne et al. (2010), we found agreement on Napp as the main factor affecting PW change in structure and a noticeable influence at the smallest scales, highlighting the importance of crop melon space arrangement.

## 4. Conclusions

Four variables, the biomass and nitrogen content of wheat and grain, have been studying on transect data selected from a set of experimental plots where different fertigation treatments were applied to a previous melon crop.

First, classical statistics were applied without considering the spatial arrangement to study these variables. None presented extreme values of kurtosis and asymmetry, but comparing the values showed a difference between variables related to nitrogen content and variables related to weight. In addition, the coefficient of variation were lower in the nitrogen-related variables.

Then, the relationships among the variables and with the nitrogen applied to the previous crop were studied. The positive effect of N addition to the melon experiment was observed through increased grain weight (GW), wheat N content (PN) and wheat weight (PW), but even these correlations present a high volatility, and it is not clear if a first- or second-order regression could fit better. However, GW versus PN and PW

versus PN presented a clear logarithmic relation tending to a maximum.

Considering the spatial arrangement of the variables' values, we have conducted a multifractal analysis on transect data as we checked that there was a non-significant trend along the transect. The Dq obtained indicates a non-strong multiscale structure in the four variables studied, but different strength was nonetheless observed between

variables related to nitrogen content (*GN* and *PN*) and variables related to weight (*GW* and *PW*). In this case, the generalized dimensions did not give us the relevant information we expected on multiscale heterogeneity but did discriminate between the two types of variables, as in the classical statistics.

A relative entropy analysis was used to identify local maxima within the data structure. Grain and plant weight (*GW* and *PW*, respectively) present a maximum structure at a scale of 5 m that corresponds to Napp treatment, as well as wheat nitrogen content (*PN*). In contrast, for the grain nitrogen content (*GN*) the maximum structure is found at 10 m, revealing that Napp is not the main factor explaining its variation. Therefore, relative entropy showed a distinction between variables related to nitrogen content that was not found using classical statistics or multifractal analysis.

The proposed approach provides information about scale dependencies related to factors that created spatial variability and is complementary to multiscale analysis and descriptive statistics.

**Acknowledgements**

This project has been partially supported by INIA-RTA04-111-C3 and by the Ministerio de Economía y Competitividad (MINECO) under Contract No. MTM2015-63914-P and No CICYT PCIN-2014-080.

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

**Table 1.** The treatments applied to the melon crop, total irrigation (applied irrigation, taking initial establishment irrigation into account, in the different treatments: 60% ETc (W1), 100% ETc (W2) and 140% ETc (W3) (15 to 104 DAT)) and applied nitrogen information. From Milne et al. (2010) with permission.

| Treatment | | Irrigation (mm) | N applied (kg N ha$^{-1}$) | | |
|---|---|---|---|---|---|
| Irrigation | Fertilizer | | Irrigation water | Fertilizer | Total |
| | N0 | | | 0 | 55.58 |
| W1 | N1 | 342.6 | 55.58 | 150 | 205.58 |
| | N2 | | | 300 | 355.58 |
| | | | | | |
| | N0 | | | 0 | 92.78 |
| W2 | N1 | 552.9 | 92.78 | 150 | 242.78 |
| | N2 | | | 300 | 392.78 |
| | | | | | |
| | N0 | | | 0 | 129.46 |
| W3 | N1 | 755.9 | 129.46 | 150 | 279.46 |
| | N2 | | | 300 | 429.46 |

**Table 2.** Descriptive Statistics of variables studied: grain N content (*GN*), grain weight (*GW*), wheat N content (*PN*) and wheat weight (*PW*).

| Statistics | *GN* | *GW* | *PN* | *PW* |
|---|---|---|---|---|
| Average | 59.01 | 5531.82 | 72.58 | 10365.20 |
| Median | 54.84 | 5404.10 | 64.82 | 10016.34 |
| Standard deviation | 28.64 | 1885.18 | 34.08 | 3604.59 |
| Variance | 820.03 | 3553897.70 | 1161.28 | 12993051.45 |
| Coefficient of variation | 0.49 | 0.34 | 0.47 | 0.35 |
| Kurtosis | 0.09 | -0.82 | -0.12 | -0.78 |
| Asymmetry | 0.80 | 0.26 | 0.76 | 0.30 |

**Table 3.** Statistical trend significance between the variables studied and distance in the transect (see Fig. 3): grain N content (*GN*), grain weight (*GW*), wheat N content (*PN*) and wheat weight (*PW*).

| | *GN* | *GW* | *PN* | *PW* |
|---|---|---|---|---|
| slope | 0,21118 | -4,34944 | 0,15982 | 1,70951 |
| s.e. | 0,11690 | 6,46473 | 0,11633 | 12,37794 |
| $R^2$ | 0,02919 | 0,00286 | 0,01180 | 0,00012 |
| t estimated | 1,07253 | 0,67279 | 1,37376 | 0,13811 |
| t value | 1,97509 | 1,97509 | 1,97509 | 1,97509 |
| significance | ns | ns | ns | ns |

# FIGURE CAPTIONS

**Fig 1.** A croquis of the experimental melon crop layout. The nine subplots of the melon crop experiment through which the wheat transect ran are shown. The wheat transect is shown by the dark green line. The fertilizer levels are shown on the figure: N0, N1, N2 and represent 0, 150 and 300 kg N ha$^{-1}$ respectively. The three different irrigation levels are indicated by the colour of the subplot lines: light blue is W1, the light green W2, and the orange W3 corresponding to 60%, 100%, and 140% of the estimated crop evapotranspiration (Ec) respectively. From different sizes subplots an example as how the melon crop are located is showed.

**Fig 2.** Monthly precipitation and irrigation applied, in mm, for melon and wheat crop.

**Fig 3.** Original data of the four variables studied including the nitrogen doses applied in the melon crop along the transect: A) Grain nitrogen content (*GN*), B) Grain weight (*GW*), C) Wheat nitrogen content (*PN*) and D) Wheat weight (*PW*). Black line represents the trend of each variable versus distance (see Table 3).

**Fig 4.** Correlations with nitrogen applied (Napp) of each variable: A) Grain nitrogen content (*GN*), B) Grain weight (*GW*), C) Wheat nitrogen content (*PN*) and D) Wheat weight (*PW*).

**Fig 5.** Effect of N applied in previous melon crop on: A) grain weight and wheat N content; B) wheat weight and wheat N content; C) grain weight and grain N content.

**Fig 6.** Multifractal analysis of the four variables studied: A) Function $\tau(q)$ versus q, B) derived generalized dimensions ($D_q$) from $\tau(q)$. The plotted variables are Grain nitrogen content (*GN*), Grain weight (*GW*), Wheat nitrogen content (*PN*) and Wheat weight (*PW*).

**Fig 7.** Entropy study: A) relative entropy, $E(\delta)$, of nitrogen applied (Napp), B) increment of relative entropy, $\Delta E(\delta)$, of Napp. The equivalent distance to the number of data points ($\delta$) are marked in $E(\delta)$.

10    **Fig 8.** Relative entropy ($E(\delta)$) respect to number of data points ($\delta$) of: A) Grain nitrogen content (*GN*), B) Grain weight (*GW*), C) Wheat nitrogen content (*PN*) and D) Wheat weight (*PW*). Black lines represents $E(\delta)$ based on entropy dimension ($D_1$) of each variable.

15    **Fig 9.** Increment of relative entropy ($\Delta E(\delta)$) respect to number of data points ($\delta$) of: A) Grain nitrogen content (*GN*), B) Grain weight (*GW*), C) Wheat nitrogen content (*PN*) and D) Wheat weight (*PW*). Black lines represents $\Delta E(\delta)$ based on entropy dimension ($D_1$) of each variable.

**Fig. 1**

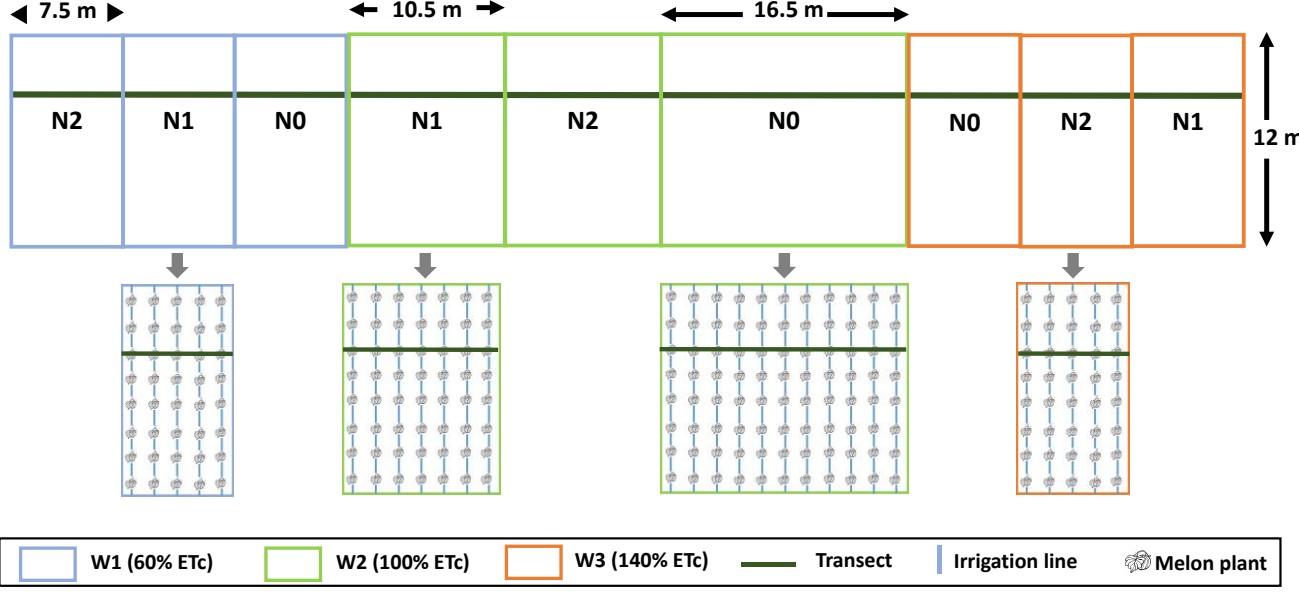

| W1 (60% ETc) | W2 (100% ETc) | W3 (140% ETc) | —— Transect | Irrigation line | Melon plant |

**Fig. 2**

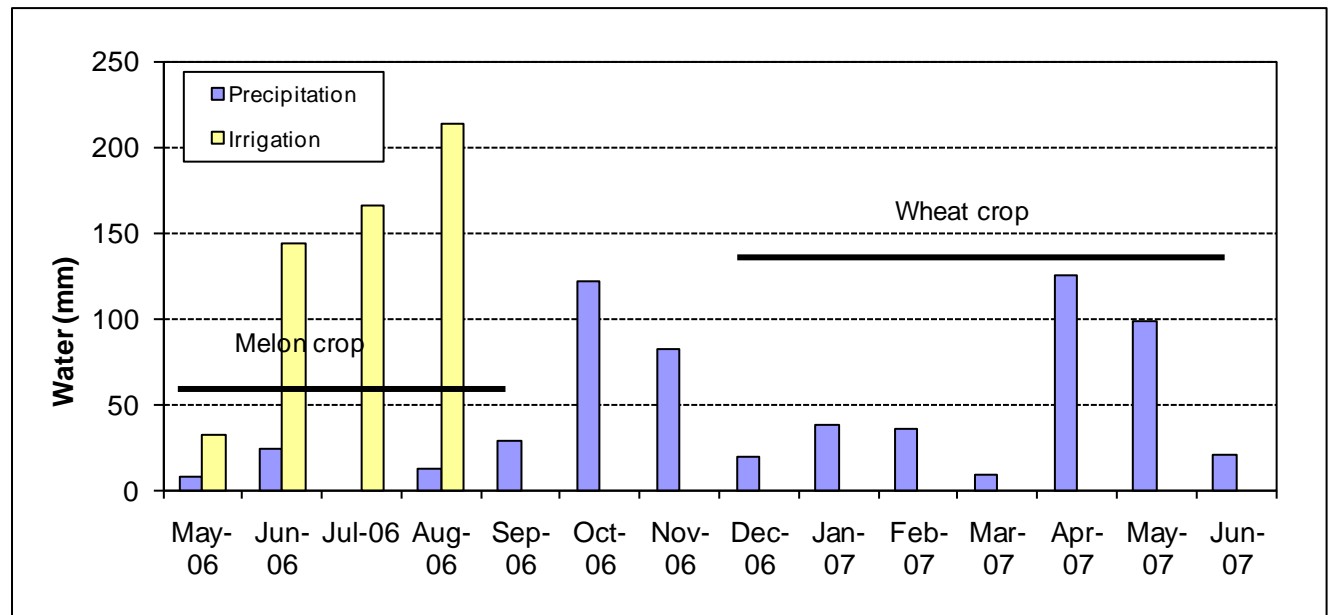

**Fig. 3**

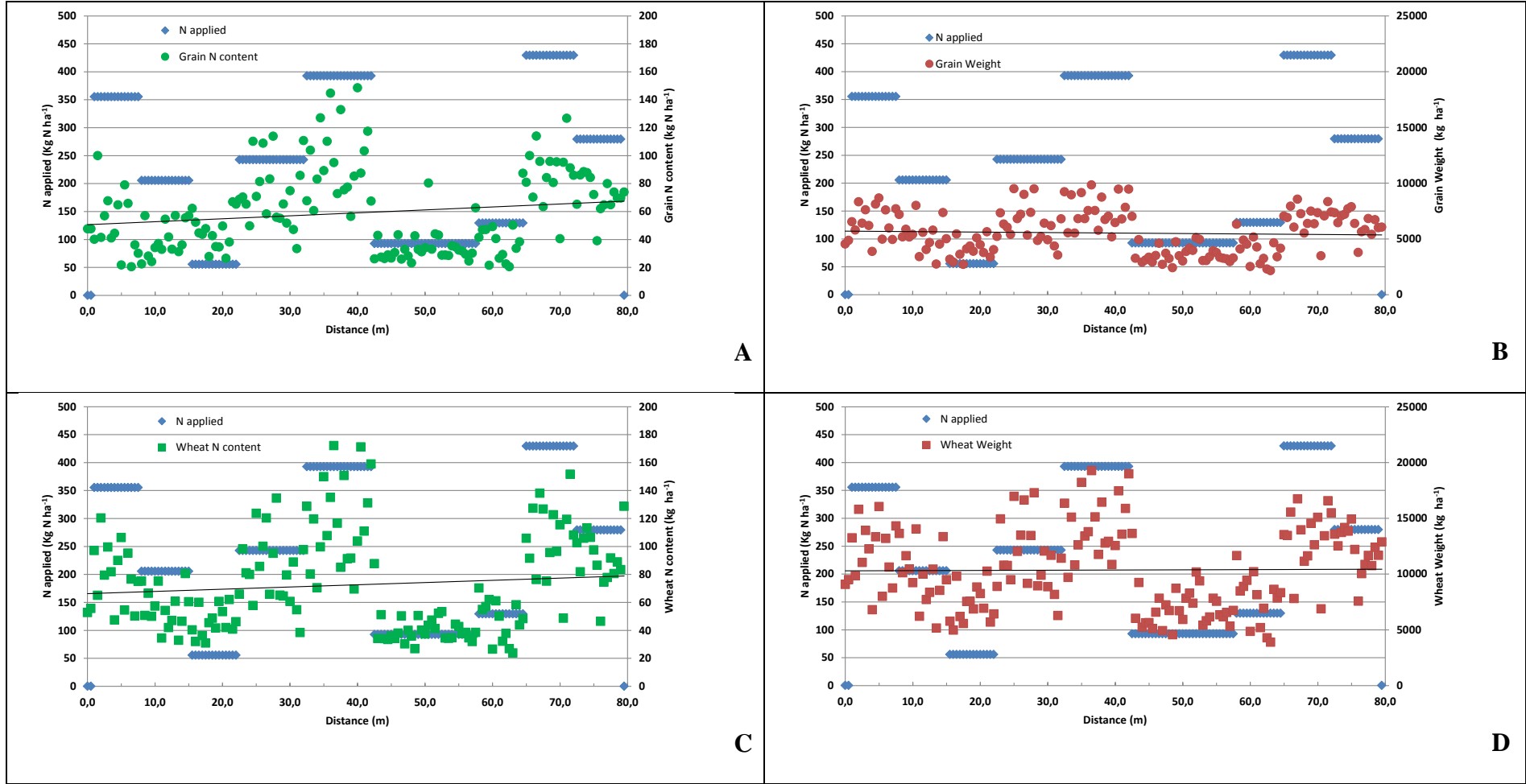

**Fig. 4**

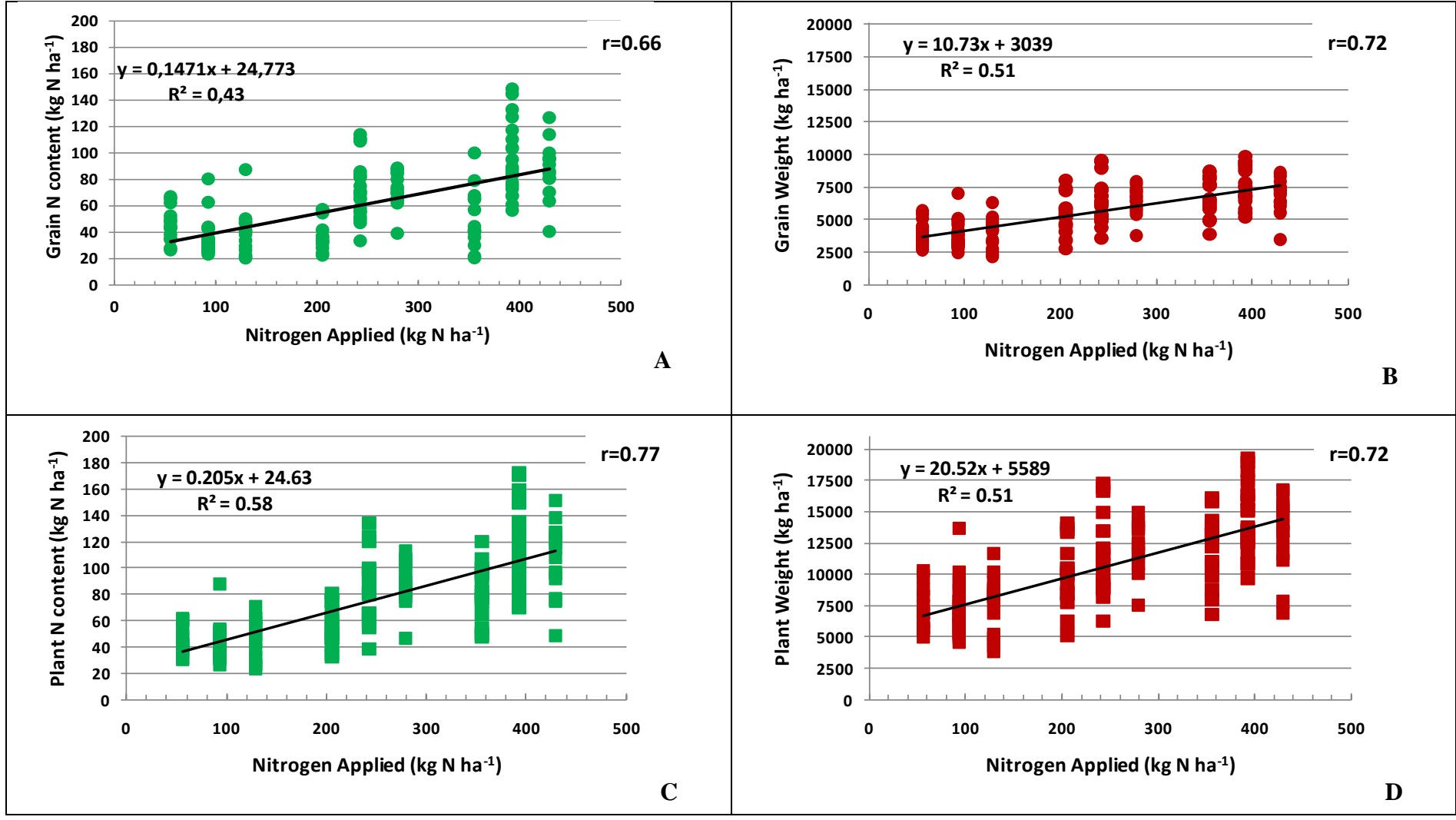

**Fig. 5**

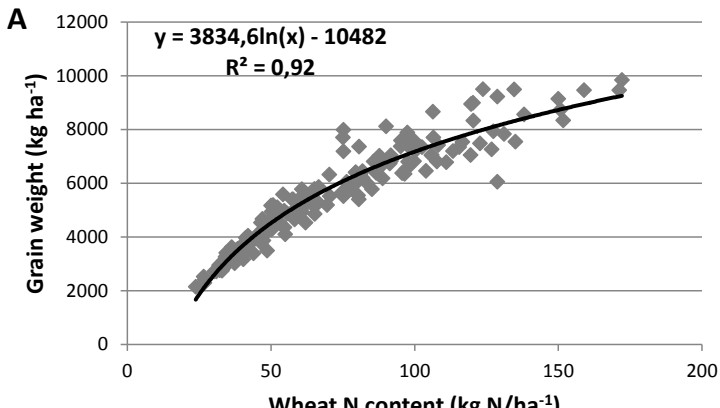

A

y = 3834,6ln(x) - 10482
R² = 0,92

Grain weight (kg ha$^{-1}$)

Wheat N content (kg N/ha$^{-1}$)

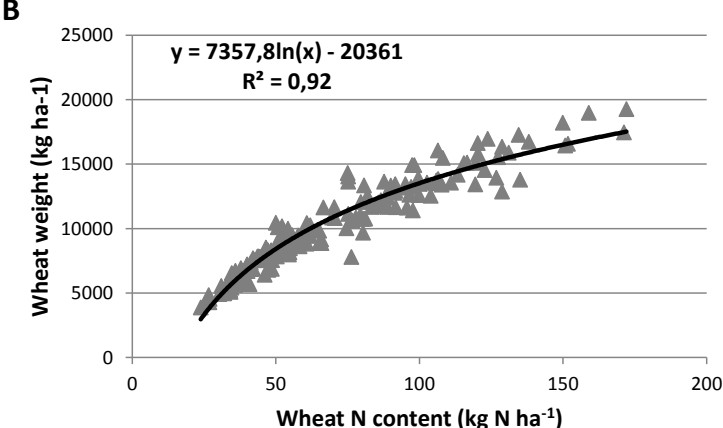

B

y = 7357,8ln(x) - 20361
R² = 0,92

Wheat weight (kg ha-1)

Wheat N content (kg N ha$^{-1}$)

**Fig 6.**

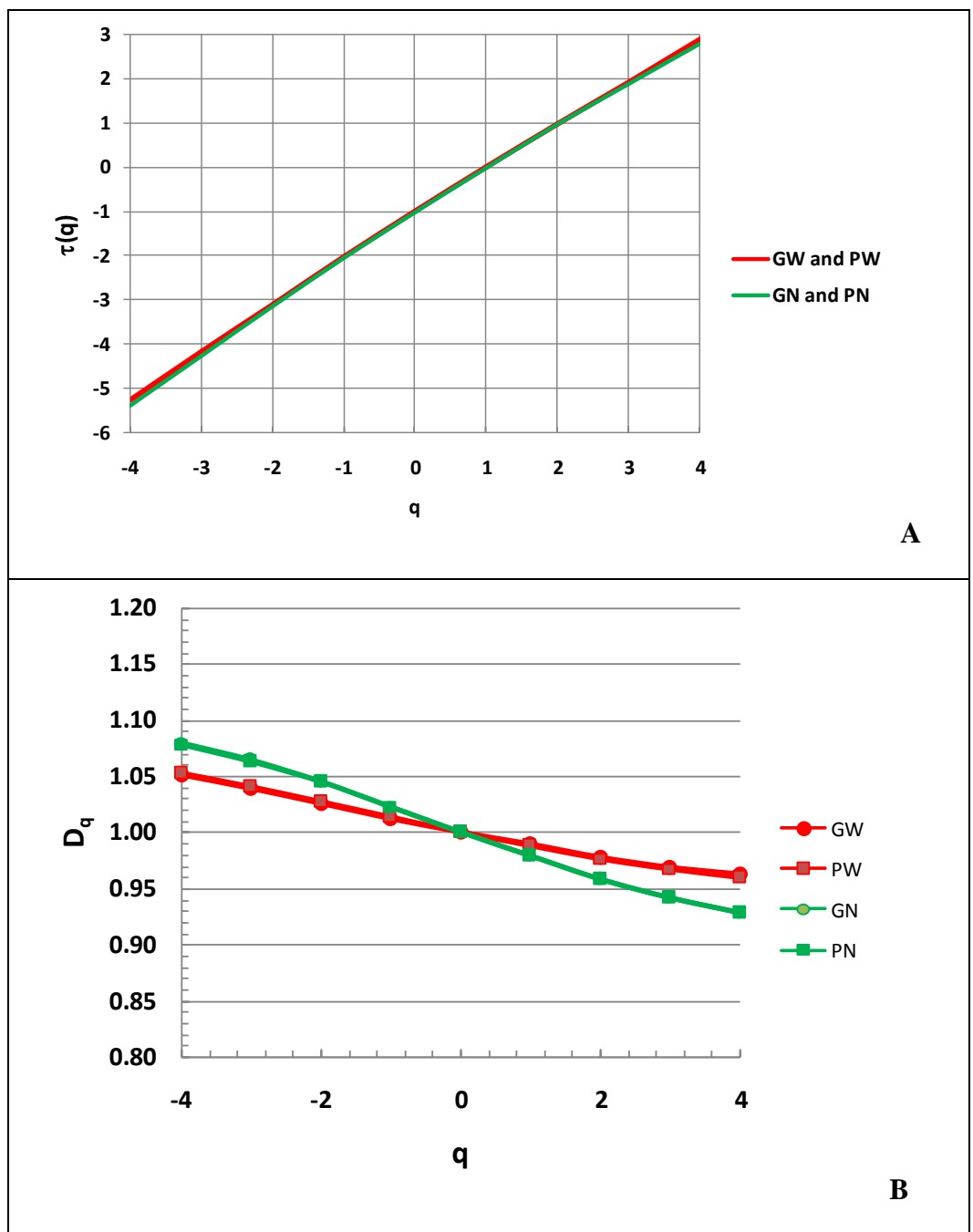

**Fig. 7**

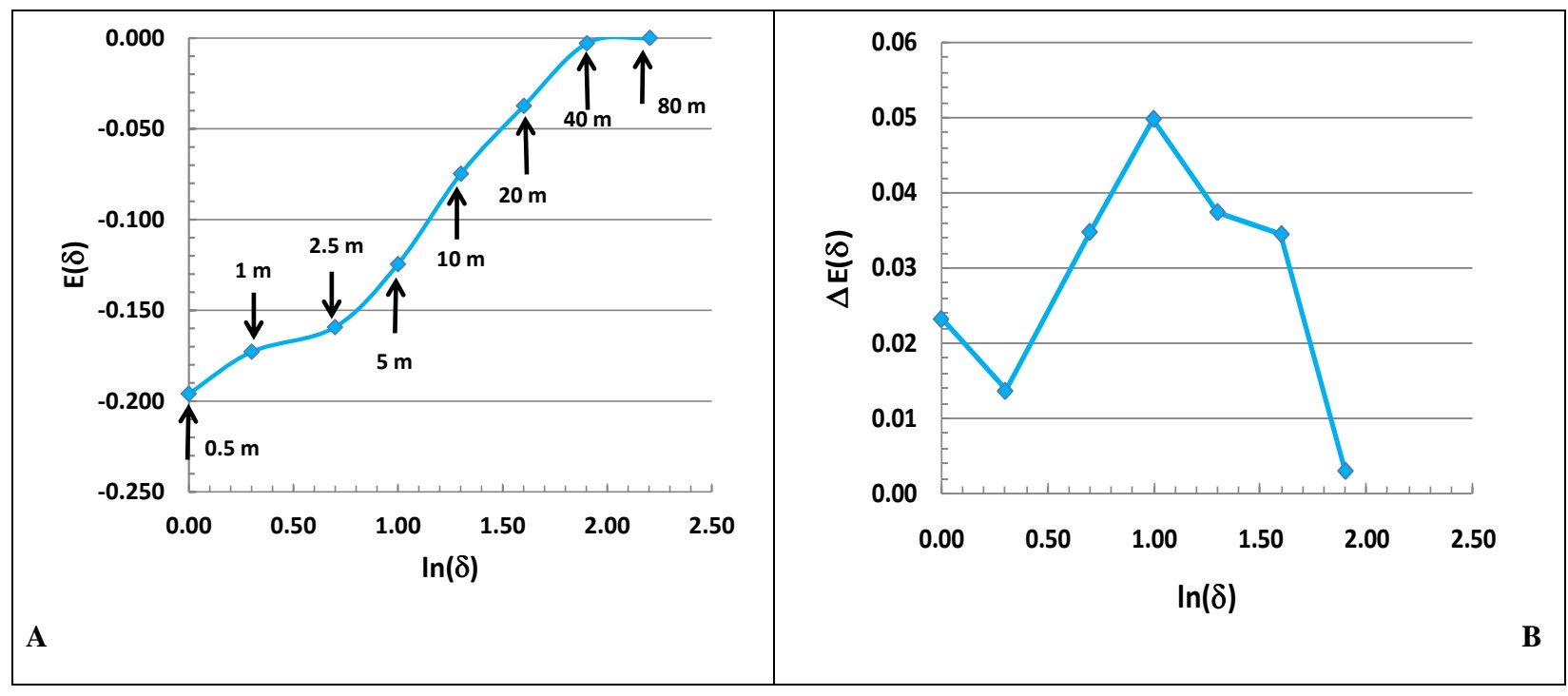

A

B

**Fig. 8**

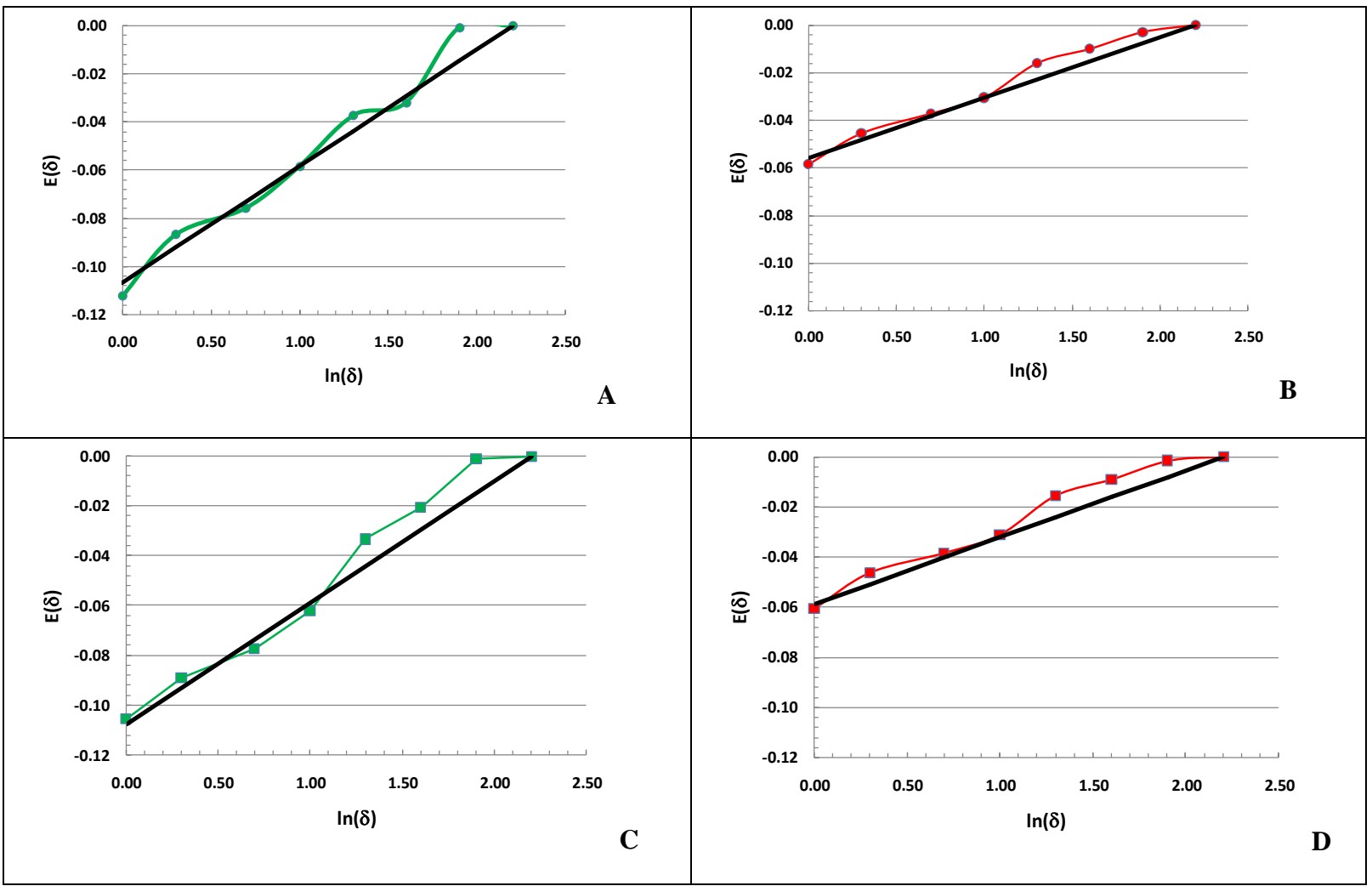

**Fig. 9**

