# Peer review of "Scale and space dependencies of soil nitrogen variability"

_Nonlinear Processes in Geophysics, 2016_

## Referee Comment (RC1) · Anonymous Referee #1 · 27 Jun 2016

The manuscript explores the effect of the N fertilizer applied to a previous horticultural crop on the subsequent, unfertilized, wheat crop: the different response of weight and nitrogen content of the cereal. The differences shown by the wheat crop after the fertilization of the previous crop were already examined by several of the authors using the wavelet technique (Milne et al. 2010). The new aspect considered in this manuscript is the separation between the whole plant and the grain. The authors discussed some results like the different answer of grain weight compared to plant weight which might be due to physiological reasons, as for instance an upper threshold for grain yield, which could be similar to what Hawkesford (2014) indicates in his figures 2 and 3.

Nevertheless the authors do not try to search for the reasons of the different behavior of the whole plant and the grain, but they show that the differences observed in their data, figures 3B and 3D of the manuscript, could be appreciated too with the multifractal

analysis using the transect sampling.

The manuscript needs a major revision: the discussions and conclusions sections do not fully agree with the abstract, the discussion section requires a clarification, as well as other sections.

Specific comments:

There are several questions:

1. Given the dry period between November2006-April 2007, seen in figure 2, and the high grain yields of figure 3, did the wheat plants receive any irrigation? In the affirmative case was the N contribution computed?

2. The data of Table 1 require some additional explanation: if the 60% of the ETc is 251.8 mm why the irrigation volume in the W1 treatment was 344.1 mm?

3. The explanations of Lines 10-18 of section 3.3, page 12 are not evident. The legend of the abscissa axes of figures 6, 7, and 8, should indicate the unit of the variable delta.

4. The use of the English language must be thoroughly revised.

Technical corrections:

Page 1, Line 2: According to Milne et al. (2010) M.C. Cartagena super-index 3 should be 4. Page 4, Line 14: The authors must indicate what UH mean. Page 4, Line 15: write 6,953 km2 and 3,192 km2. Page 4, Line 16: delete 'caliche'. Page 4, Lines 19-22: rewrite the two sentences. Page 5, Lines 12-13: the soil could belong to the xeralf suborder, and might have a petrocalcic horizon, but it does not necessarily mean that the soil can be classified as written in the manuscript. Page 5, Line 16: the proper units are molc m-3. Page 6, Line 1: if the plant density for wheat is written in plants m-2 in page 7 line , why do not use similar units here: 4.44 plants m-2? Page 6, Line 8: what does DAT stand for? Page 8, Line 18: write 'The probability is' instead of 'We now perform a weighted sum over all segments that yield to'. Page 17, Line 1: insert

the reference Soil Survey Staff 1999 Table 1: is it necessary? Table 1: the question might be irrelevant but why the numbers are not equal to those of Table 1 of Milne et al. (2010)? Table 1: if the Table is kept in the manuscript the third, fourth, sixth, and ninth columns could be deleted. The relevant information could be reduced to the ETo, kc, and rain depth data.

Reference:

Hawkesford, M.J. 2010. Reducing the reliance on nitrogen fertilizer for wheat production. J. Cereal Sci. 59:276-283.

---

## Referee Comment (RC2) · Anonymous Referee #2 · 21 Jul 2016

Overall, it is an interesting work that addresses scale-dependence of structure in series. Three components of the work require better explanations. 1. The transect crosses areas with different treatments. This is reflected in responses to N shown in Fig. 3. The multifractal formalism does not allow for trends. How then the deterministic component of variation is reflected in multifractal parameters? 2. Distances of 5 and 10 m are mentioned as the distances at which structure is best revealed. Why the numbers are round? What is the method of finding these numbers? Do these numbers depend on the spatial increment of measurements? 3. Authors are talking about structure throughout the manuscript. But what is structure? How is it defined? It is important for future attempts to relate structure and function.

The manuscript requires editing for English. There are many small pesky errors. Here are examples from first two pages. Page 2 9 Change "can be seen as the result of" to

"exhibit" 14 Change "Logsdom" to "Logsdon" 18 Change "on a" to "in" 20 Change "the scaling property" to "scaling propertirs" Page 3 5 Change "in" to "to" 9 "common"? 12 How a surface site can be located near an aquifer? Wat are you trying to say with this characterization? 15 levels 21 "Nitrogen" not capital

---

## Referee Comment (RC3) · Anonymous Referee #3 · 28 Jul 2016

The manuscript deals with the effect of residual soil N content, resulting from a previous experiment with melon, on several parameters in a wheat crop, including grain and plant N content and biomass. The main objective was to identify the structure of the variations in these parameters along a transect at different scales, for which the authors apply multifractal and entropy analyses. The topic of this work is interesting for a wide range of potential readers, and the analyses conducted, although previously used for other parameters, are novel when considering the crop parameters covered. However, my recommendation on the manuscript is that it needs a major revision for a series of reasons:

-The introduction section is not well constructed, and contains some paragraphs (more precisely, P. 3, L. 12-20) that are a mere description of the experimental setup. This description should be part of the Material and Methods section and not the Introduction.

Moreover, since several other papers with data from this experiment have been published already, their main findings should be included in this section (e.g., Castellanos et al., 2010; Milne et al., 2010).

-The Material and Methods section includes a detailed description of a previous experiment with melon plants that was conducted prior to the establishment of the wheat crop. Although knowing the history of the plots is necessary for the interpretation of the data, many of the details that the authors include are not relevant for the present work, since only parameters of wheat are discussed. For example, melon plant density (P.4, L. 14-15) or the number or rows and plants per row (P.4, L.17), or the details of melon plants (P. 4, L 12-13) are just irrelevant information. The information on the melon experiment should be revised and only the aspects that are important to understand the wheat data should be kept (fertilization, irrigation, and similar). Also, Figure 1 indicates the plot distribution for the different treatments in the melon experiment, when only the upper line of plots, which are the ones crossed by the transect, are needed in this paper. The figure should be revised to remove unnecessary information.

-The results and discussion section is very limited (roughly, one page in length). In my opinion, the authors should do a better job describing and specially discussing the results and the implications of their findings. For example, Milne et al. (2010) used the same data reported here but subjected to a different type of analysis. I might suggest comparing both analyses and discuss differences and similarities. Also, the authors could discuss other aspects shown by the data, as why wheat grain weight does not increase substantially with N applications above approximately 150 kg/ha, while N content increases both in the plant and in the grain and plant biomass increases with increasing N. -The English of the text should be the subject of a deep revision. There are many mistakes and colloquial expressions that should be removed.

Some specific comments:

The text and expressions should be revised. For example, P.3, L.4 "This can give us an

insight into the dominant processes". This sentence seems unfinished (processes governing something?). As another example, in P.3, L. 5-11: the word "scale" is repeated too many times "to study scale effects localized in scale".

In P. 3, L 20. What the authors did was to analyze the differences in some plant parameters that may be caused by residual N. However, residual soil N is not evaluated in this work, and the procedures used do not allow to do that. Therefore, this sentence should be deleted.

-Do you, by any chance, have any numbers about N exports from the plots in the melon experiments? This could be very valuable information in order to understand the starting point of the wheat experiment.

-Revise the Soil Taxonomy classification of this soil (P.4, L.4).

-Check the separators used for decimals and thousands (e.g., P.4, L.6 and 7: "7,9", "2,2"). -P.4,L.12. "The species…" replace with "The variety…". In the same line, "Cucumismelo" should be replaced by "Cucumis melo".

-Table 1 and figure 4. The N-application treatments in the melon experiment are only three, but in figure 4 there are 9 application rates. I guess that this is due to the addition of different irrigation amounts to the plots, which contain some amount of N. These amounts are not indicated in table 1 clearly, probably due to some mistake when preparing the table. I understand from Milne et al. (2010) that it should be the third column from the right in this table.

In figure 4, and considering the high variability that the treatments present, it might be necessary to calculate the confidence interval for the slope of the regression lines. It seems to me that in the Grain weight vs. N applied the 0 will be included in this interval, and thus no linear relation could be

Overall, the manuscript needs a deep revision prior to be accepted for publication in Non-linear Processes in Geophysics.

---

## Author Comment (AC1) · 27 Nov 2016

The manuscript explores the effect of the N fertilizer applied to a previous horticultural crop on the subsequent, unfertilized, wheat crop: the different response of weight and nitrogen content of the cereal. The differences shown by the wheat crop after the fertilization of the previous crop were already examined by several of the authors using the wavelet technique (Milne et al. 2010). The new aspect considered in this manuscript is the separation between the whole plant and the grain. The authors discussed some results like the different answer of grain weight compared to plant weight which might be due to physiological reasons, as for instance an upper threshold for grain yield, which could be similar to what Hawkesford (2014) indicates in his figures 2 and 3. Thank you very much for your comments. At Milne et al. (2010) the work was centered in plant weight (wheat weight or PW) and in this manuscript we study plant weight (PW), plant

Nitrogen content (PN), grain weight (wheat yield or GW) and grain Nitrogen content (GN). Thank you for the reference of Hawkesford (2014) that we have included in this work.

Nevertheless the authors do not try to search for the reasons of the different behavior of the whole plant and the grain, but they show that the differences observed in their data, figures 3B and 3D of the manuscript, could be appreciated too with the multifractal analysis using the transect sampling. We wanted to apply a multifractal analysis and the relative entropy to compare the behaviour of these four variables. However, we have included the relations between variables to improve the discussion in section 3.1: The positive effect of increasing grain weight together with the additional benefit of increasing wheat N content with increasing N application is shown in Fig. 5A. Moreover, the same positive effect of N addition was observed, increasing wheat weight together with increasing wheat N content (Fig. 5B). Closer inspection of Fig. 4 reveals that the variability was much higher when the N application was higher. Barraclough et al. (2010), in an experiment with N fertilization applied homogenously directly to the wheat crop, found that much of the additional N taken up by the plant (PN) is manifested in higher yield (GW), although we remark again that in this work, the N application was performed in the melon crop experiment, through fertigation on crop lines, and the wheat crop did not receive any N fertilization and was not irrigated. This positive effect of N addition has been observed in numerous studies (Barraclough et al., 2010 and references therein). Several works determine the N optimum in the wheat crop, but in this study, the optimal N dose was not obtained because we sought to study the variability and the effect of the residual N resulting from N application to a previous melon crop months before.

Fig 5. Effect of N applied in previous melon crop on: A) grain weight and wheat N content; B) wheat weight and wheat N content; C) grain weight and grain N content.

Fig 7. Entropy study: A) relative entropy, E(ïĄď), of Nitrogen applied (Napp), B) increment of relative entropy, ïĄĎE(ïĄď), of Napp. The equivalent distance to the number

of data points (ïĄď) are marked in E(ïĄď).

======= The manuscript needs a major revision: the discussions and conclusions sections do not fully agree with the abstract, the discussion section requires a clarification, as well as other sections. We have improved the discussion and conclusions sections as there were some mistakes.

Specific comments. There are several questions: 1. Given the dry period between November2006-April 2007, seen in figure 2, and the high grain yields of figure 3, did the wheat plants receive any irrigation? In the affirmative case was the N contribution computed? No, the plants did not receive any irrigation. The yields were ranged between 3.7 and 7.5 t/ha following the Ministery of Agriculture statistics data.

2. The data of Table 1 require some additional explanation: if the 60% of the ETc is 251.8 mm why the irrigation volume in the W1 treatment was 344.1 mm? We have included the explication to this in the text. The rainfall was negligible, so the water applied was calculated as the ratio between the ETc of the previous week and the efficiency of the system, which considers the salt tolerance of the crop, the quality of the irrigation, soil texture and the homogeneity of the irrigation system (Rincón and Giménez (1989)), estimated as 0.81 under the study conditions. This result, called theoretical irrigation (irrigation calculated), was divided by the number of days to obtain the daily irrigation requirements. The real irrigation was the amount of water registered on the water meter (irrigation applied).

Rincón, L., Giménez, M., 1989. Fertirrigación por goteo en melón. Fertilización 105, 55–56.

3. The explanations of Lines 10-18 of section 3.3, page 12 are not evident. The legend of the abscissa axes of figures 6, 7, and 8, should indicate the unit of the variable delta.

The figures mentioned are now are 7, 8 and 9 plot. We have now improved the captions of these figures clarifying that "ïĄď" is the number of data points used and in the first
figure (figure 7) has been translated into meters so the reader can follow the results better.

========

Fig 8. Relative entropy (E(ïĄď)) respect to number of data points (ïĄď) of: A) Grain Nitrogen content (GN), B) Grain Weight (GW), C) Wheat Nitrogen content (PN) and D) Wheat Weight (PW). Black lines represents E(ïĄď) based on entropy dimension (D1) of each variable.

Fig 9. Increment of relative entropy (ïĄĎE(ïĄď)) respect to number of data points (ïĄď) of: A) Grain Nitrogen content (GN), B) Grain Weight (GW), C) Wheat Nitrogen content (PN) and D) Wheat Weight (PW). Black lines represents ïĄĎE(ïĄď) based on entropy dimension (D1) of each variable.

Also we have clarified more the text: The increments of the E(ïĄď) (ïĄĎE(ïĄď)), between two consecutives scales, calculated for Napp and the four variables are shown in Fig. 7B and Fig. 9, respectively. PN, GW and PW present a similar scaling trend, with a maximum structure revealed at scale ïĄď=10, corresponding to a distance of 5 m. This behaviour is the same found in Napp in the melon crop. In the case of GN, the maximum structure is found at ïĄď=20 (10 m), indicating that the interaction of other factors influences in this variation, and the Napp is not the main one. All the values of ïĄĎE(ïĄď) at the smallest scales, ïĄď=5, 2 and 1 (2.5, 1 and 0.5 m respectively), show an increase, giving the second maximum value for GN, GW and PW. This result suggests that at those scales, the variation is mainly due to the melon cropping lines, as the uptake of the applied nitrogen by this crop left a lower amount of available nitrogen for the wheat crop. In the case of PN, the second maximum was found at ïĄď=20 (10 m) followed by the one at the smallest scales, ïĄď=2 and 1 (1 and 0.5 m), as in the other variables.

4. The use of the English language must be thoroughly revised. It has been revised and a certificate of the translator is included.

Technical corrections: Page 1, Line 2: According to Milne et al. (2010) M.C. Cartagena super-index 3 should be 4. Done. Page 4, Line 14: The authors must indicate what UH mean. Done. Page 4, Line 15: write 6,953 km2 and 3,192 km2. Done. Page 4, Line 16: delete 'caliche'. Done. Page 4, Lines 19-22: rewrite the two sentences. Done. Page 5, Lines 12-13: the soil could belong to the xeralf suborder, and might have a petrocalcic horizon, but it does not necessarily mean that the soil can be classified as written in the manuscript. We are sorry; there was a mistake in the classification of the soil. We have corrected it. Page 6, Line 1: if the plant density for wheat is written in plants m-2 in page 7 line , why do not use similar units here: 4.44 plants m-2? Well, the density to melon crop is 0.444 plans m-2, so this unit is not used very much. Page 6, Line 8: what does DAT stand for? We have removed DAT in all the paper. Page 8, Line 18: write 'The probability is' instead of 'We now perform a weighted sum over all segments that yield to' Done. Page 17, Line 1: insert the reference Soil Survey Staff 1999 Done. Table 1: is it necessary? Table 1: the question might be irrelevant but why the numbers are not equal to those of Table 1 of Milne et al. (2010)? Table 1: if the Table is kept in the manuscript the third, fourth, sixth, and ninth columns could be deleted. The relevant information could be reduced to the ETo, kc, and rain depth data. We have removed the indicated columns and have corrected the mistakes. The nine columns have not been removed because the referee 3 did not understand the N treatments, so the nine columns is necessary to clarify the N treatments.

Table 1. The treatments applied to the melon crop, total irrigation (applied irrigation, taking initial establishment irrigation into account, in the different treatments: 60% ETc (W1), 100% ETc (W2) and 140% ETc (W3) (15 to 104 DAT)) and applied nitrogen information. From Milne et al. (2010) with permission.

========================

Reference: Hawkesford, M.J. 2010. Reducing the reliance on nitrogen fertilizer for wheat production. J. Cereal Sci. 59:276-283. Included now in the manuscript.

Please also note the supplement to this comment:
http://www.nonlin-processes-geophys-discuss.net/npg-2016-32/npg-2016-32-AC1-supplement.pdf
* * *
Interactive
comment

[Figure]

Interactive
comment

[revised manuscript text omitted]

A

B

**Fig. 8**

[Figure]

[Figure]

A

B

C

D

---

## Author Comment (AC2) · 27 Nov 2016

Anonymous Referee #2 Overall, it is an interesting work that addresses scale-dependence of structure in series. Thank you for your comment.

Three components of the work require better explanations: 1. The transect crosses areas with different treatments. This is reflected in responses to N shown in Fig. 3. The multifractal formalism does not allow for trends. How then the deterministic component of variation is reflected in multifractal parameters?

Figure 4 is showing the relation of nitrogen applied in melon crop and the values of the four variables study and of course that there is a relation. But to study the tendency in the transect for each variable we have to study Figure 3. For that we have done a statistical test to see if the slope of the data versus distance has a significant value or

not.

At the end of section 2.4: Finally, a statistical test was applied for each variable to determine if there was any significant trend with distance that would not allow the application of a straight multifractal analysis on the original data. The measure used was the coefficient of the slope of the regression line along the distance. This coefficient is derived using the least squares method and then compared to zero using the Student t-test. If the t value is less than a critical t value at the 95% level for the degrees of freedom, then the slope is considered to be zero.

At the end of section 3.1: Before applying the multifractal analysis, a statistical test was applied to each variable to determine whether it presented a significant trend with distance. The results are shown in Table 3, where the estimated t was always lower than the critical t-value, implying that no spatial trend was significant.

We have included a new table: (see Table 3)

2. Distances of 5 and 10 m are mentioned as the distances at which structure is best revealed. Why the numbers are round? What is the method of finding these numbers? Do these numbers depend on the spatial increment of measurements? The data were obtained each 0.5 m. The relation between number of data points and equivalent distance is added in Figure 7.

3. Authors are talking about structure throughout the manuscript. But what is structure? How is it defined? It is important for future attempts to relate structure and function. At the introduction we have added: Geostatistical methods and, more recently, multifractal/wavelet techniques have been used to characterize the scaling and heterogeneity of soil properties, among other approaches coming from complexity science (de Bartolo et al., 2011). These methods study the structure of the property measured in the sense that compares the probability distribution at each scale and among scales.

The manuscript requires editing for English. It has been revised and a certificate of the

translator is included.

There are many small pesky errors. Here are examples from first two pages. Page 2 9 Change "can be seen as the result of" to "exhibit". Done. Change "Logsdom" to "Logsdon". Done. Change "on a" to "in" 20 Change "the scaling property" to "scaling propertirs". Done. How a surface site can be located near an aquifer? The irrigated agriculture is an activity very important in this area and principally is irrigated agriculture, which is located near to groundwater sources. Mancha Occidental aquifer and Campo de Montiel Aquifer are the main sources of water in more than the half-irrigated lands (Domínguez and de Juan, 2008).

Domínguez, A., J.A. de Juan. 2008. Agricultural water management in Castilla-La Mancha (Spain). p. 69-128. In: Agricultural water management Research Trends, Magnus L. Sorensen (Ed.). Nova Science Publishers, New York.

What are you trying to say with this characterization? We are describing the importance of water and nitrogen in this area with special characteristics in the soil and type of crops.

"Nitrogen" not capital. Done.

Please also note the supplement to this comment:
http://www.nonlin-processes-geophys-discuss.net/npg-2016-32/npg-2016-32-AC2-supplement.zip
* * *
[Figure]

**Table 3.** Statistical trend significance between the variables studied and distance in the transect (see Fig. 3): grain N content (*GN*), grain weight (*GW*), wheat N content (*PN*) and wheat weight (*PW*).

| | *GN* | *GW* | *PN* | *PW* |
|---|---|---|---|---|
| slope | 0.21118 | -4.34944 | 0.15982 | 1.70951 |
| s.e. | 0.11690 | 6.46473 | 0.11633 | 12.37794 |
| $R^2$ | 0.02919 | 0.00286 | 0.01180 | 0.00012 |
| t estimated | 1.07253 | 0.67279 | 1.37376 | 0.13811 |
| t value | 1.97509 | 1.97509 | 1.97509 | 1.97509 |
| significance | ns | ns | ns | ns |

[Figure]

**Fig 7.** Entropy study: A) relative entropy, E(δ), of Nitrogen applied (Napp), B) increment of relative entropy, ΔE(δ), of Napp. The equivalent distance to the number of data points (δ) are marked in E(δ).

---

## Author Comment (AC3) · 27 Nov 2016

Anonymous Referee #3 The manuscript deals with the effect of residual soil N content, resulting from a previous experiment with melon, on several parameters in a wheat crop, including grain and plant N content and biomass. The main objective was to identify the structure of the variations in these parameters along a transect at different scales, for which the authors apply multifractal and entropy analyses. The topic of this work is interesting for a wide range of potential readers, and the analyses conducted, although previously used for other parameters, are novel when considering the crop parameters covered. Thank you for your comments.

However, my recommendation on the manuscript is that it needs a major revision for a series of reasons: -The introduction section is not well constructed, and contains some paragraphs (more precisely, P. 3, L. 12-20) that are a mere description of the experimental setup. This description should be part of the Material and Methods section and not the Introduction. Moreover, since several other papers with data from this experiment have been published already, their main findings should be included in this section (e.g., Castellanos et al., 2010; Milne et al., 2010). We have changed the Introduction section leaving a paragraph describing the importance of water and nitrogen in the area.

-The Material and Methods section includes a detailed description of a previous experiment with melon plants that was conducted prior to the establishment of the wheat crop. Although knowing the history of the plots is necessary for the interpretation of the data, many of the details that the authors include are not relevant for the present work, since only parameters of wheat are discussed. For example, melon plant density (P.4, L. 14-15) or the number or rows and plants per row (P.4, L.17), or the details of melon plants (P. 4, L 12-13) are just irrelevant information. The information on the melon experiment should be revised and only the aspects that are important to understand the wheat data should be kept (fertilization, irrigation, and similar). We have shorted the section on melon crop and wheat crop focusing only in the points necessary to understand the results.

Also, Figure 1 indicates the plot distribution for the different treatments in the melon experiment, when only the upper line of plots, which are the ones crossed by the transect, are needed in this paper. The figure should be revised to remove unnecessary information. We have changed Figure 1.

-The results and discussion section is very limited (roughly, one page in length). In my opinion, the authors should do a better job describing and specially discussing the results and the implications of their findings. We have improve the Discussion section remarking our findings

For example, Milne et al. (2010) used the same data reported here but subjected to a different type of analysis. I might suggest comparing both analyses and discuss differences and similarities. We have added in section 3.3: The increments of the E(ïĄď) (ïĄĎE(ïĄď)), between two consecutives scales, calculated for Napp and the four variables are shown in Fig. 7B and Fig. 9, respectively. PN, GW and PW present a similar scaling trend, with a maximum structure revealed at scale ïĄď=10, corresponding to a distance of 5 m. This behaviour is the same found in Napp in the melon crop. In the case of GN, the maximum structure is found at ïĄď=20 (10 m), indicating that the interaction of other factors influences in this variation, and the Napp is not the main one. All the values of ïĄĎE(ïĄď) at the smallest scales, ïĄď=5, 2 and 1 (2.5, 1 and 0.5 m respectively), show an increase, giving the second maximum value for GN, GW and PW. This result suggests that at those scales, the variation is mainly due to the melon cropping lines, as the uptake of the applied nitrogen by this crop left a lower amount of available nitrogen for the wheat crop. In the case of PN, the second maximum was found at ïĄď=20 (10 m) followed by the one at the smallest scales, ïĄď=2 and 1 (1 and 0.5 m), as in the other variables. Comparing these results with those published by Milne et al. (2010), we found agreement on Napp as the main factor affecting PW change in structure and a noticeable influence at the smallest scales, highlighting the importance of crop melon space arrangement.

Also, the authors could discuss other aspects shown by the data, as why wheat grain weight does not increase substantially with N applications above approximately 150 kg/ha, while N content increases both in the plant and in the grain and plant biomass increases with increasing N. We have added the following analysis in section 3.1: Classical statistical analyses were performed on each of the variables to study their first statistical moments (Table 2). We could observe that the average and median present differences for each variable, in contrast to a normal distribution where both coincide. However, kurtosis and asymmetry do not present values higher than the unit in absolute terms. GW and PW present the highest kurtosis (0.82 and 0.78) and are negative. On the other hand, GN and PN have the highest asymmetry and are positive. The
coefficient of variation is higher in variables related to nitrogen content (GN and PN) and lower in variables related to weight (GW and PW).

Also we have included results and discussion of the relation between the variables: The positive effect of increasing grain weight together with the additional benefit of increasing wheat N content with increasing N application is shown in Fig. 5A. Moreover, the same positive effect of N addition was observed, increasing wheat weight together with increasing wheat N content (Fig. 5B). Closer inspection of Fig. 4 reveals that the variability was much higher when the N application was higher. Barraclough et al. (2010), in an experiment with N fertilization applied homogenously directly to the wheat crop, found that much of the additional N taken up by the plant (PN) is manifested in higher yield (GW), although we remark again that in this work, the N application was performed in the melon crop experiment, through fertigation on crop lines, and the wheat crop did not receive any N fertilization and was not irrigated. This positive effect of N addition has been observed in numerous studies (Barraclough et al., 2010 and references therein). Several works determine the N optimum in the wheat crop, but in this study, the optimal N dose was not obtained because we sought to study the variability and the effect of the residual N resulting from N application to a previous melon crop months before.

-The English of the text should be the subject of a deep revision. There are many mistakes and colloquial expressions that should be removed. It has been revised and a certificate of the translator is included.

Some specific comments: The text and expressions should be revised. For example, P.3, L.4 "This can give us an insight into the dominant processes". This sentence seems unfinished (processes governing something?). As another example, in P.3, L. 5-11: the word "scale" is repeated too many times "to study scale effects localized in scale". We have reviewed the text to improve it.

In P. 3, L 20. What the authors did was to analyze the differences in some plant

parameters that may be caused by residual N. However, residual soil N is not evaluated in this work, and the procedures used do not allow to do that. Therefore, this sentence should be deleted. Done.

-Do you, by any chance, have any numbers about N exports from the plots in the melon experiments? This could be very valuable information in order to understand the starting point of the wheat experiment. We are really sorry but we haven't.

-Revise the Soil Taxonomy classification of this soil (P.4, L.4). Done. -Check the separators used for decimals and thousands (e.g., P.4, L.6 and 7: "7,9", "2,2"). Done. -P.4,L.12. "The species: : :" replace with "The variety: : :". Done. In the same line, "Cucumismelo" should be replaced by "Cucumis melo". Done.

-Table 1 and figure 4. The N-application treatments in the melon experiment are only three, but in figure 4 there are 9 application rates. I guess that this is due to the addition of different irrigation amounts to the plots, which contain some amount of N. These amounts are not indicated in table 1 clearly, probably due to some mistake when preparing the table. I understand from Milne et al. (2010) that it should be the third column from the right in this table. Table 1 has been changed.

In figure 4, and considering the high variability that the treatments present, it might be necessary to calculate the confidence interval for the slope of the regression lines. It seems to me that in the Grain weight vs. N applied the 0 will be included in this interval, and thus no linear relation could be. We have included the follow paragraph in section 3.1: To study the relationships of GW, PW, GN and PN with the nitrogen applied during the melon crop season (Napp), we have plotted these variables without considering any spatial factors (Fig. 4). All of them show a tendency, as we expected, to increase in value as Napp increases. The correlation coefficient (r) for the four variables range from 0.66 (GN case) up to 0.77 (PN case) demonstrating that there are statistically significant correlations with the N application in the melon crop experiment (Napp), as the wheat crop did not receive any N directly. For this reason, the relationship that we

can observe could be considered linear, as the range we are studying is suboptimal and not as in other studies (e.g., Hawkesford, 2014). However, a quadratic relation can be fitted to all the variables with a similar R2 (results not shown).

Overall, the manuscript needs a deep revision prior to be accepted for publication in Non-linear Processes in Geophysics. We have worked hard to achieve the quality required by the journal.

Please also note the supplement to this comment:
http://www.nonlin-processes-geophys-discuss.net/npg-2016-32/npg-2016-32-AC3-supplement.zip
* * *
[Figure]

**Fig 1.** A croquis of the experimental melon crop layout. The nine subplots of the melon crop experiment through which the wheat transect ran are shown. The wheat transect is shown by the dark green line. The fertilizer levels are shown on the figure: N0, N1, N2 and represent 0, 150 and 300 kg N ha$^{-1}$ respectively. The three different irrigation levels are indicated by the colour of the subplot lines: light blue is W1, the light green W2, and the orange W3 corresponding to 60%, 100%, and 140% of the estimated crop evapotranspiration (Ec) respectively. From different sizes subplots an example as how the melon crop are located is showed.

**Table 2.** Descriptive Statistics of variables studied: grain N content (*GN*), grain weight (*GW*), wheat N content (*PN*) and wheat weight (*PW*).

| Statistics | GN | GW | PN | PW |
| --- | --- | --- | --- | --- |
| Average | 59.01 | 5531.82 | 72.58 | 10365.20 |
| Median | 54.84 | 5404.10 | 64.82 | 10016.34 |
| Standard deviation | 28.64 | 1885.18 | 34.08 | 3604.59 |
| Variance | 820.03 | 3553897.70 | 1161.28 | 12993051.45 |
| Coefficient of variation | 0.49 | 0.34 | 0.47 | 0.35 |
| Kurtosis | 0.09 | -0.82 | -0.12 | -0.78 |
| Asymmetry | 0.80 | 0.26 | 0.76 | 0.30 |

**Fig 5.** Effect of N applied in previous melon crop on: A) grain weight and wheat N content; B) wheat weight and wheat N content; C) grain weight and grain N content.

**A**

[Figure]

$y = 3834{,}6\ln(x) - 10482$
$R^2 = 0{,}92$

**B**

[Figure]

$y = 7357{,}8\ln(x) - 20361$
$R^2 = 0{,}92$

**Table 1.** The treatments applied to the melon crop, total irrigation (applied irrigation, taking initial establishment irrigation into account, in the different treatments: 60% ETc (W1), 100% ETc (W2) and 140% ETc (W3) (15 to 104 DAT)) and applied nitrogen information. From Milne et al. (2010) with permission.

| Treatment | | Irrigation (mm) | N applied (kg N ha$^{-1}$) | | |
|---|---|---|---|---|---|
| Irrigation | Fertilizer | | Irrigation water | Fertilizer | Total |
| | N0 | | | 0 | 55.58 |
| W1 | N1 | 342.6 | 55.58 | 150 | 205.58 |
| | N2 | | | 300 | 355.58 |
| | | | | | |
| | N0 | | | 0 | 92.78 |
| W2 | N1 | 552.9 | 92.78 | 150 | 242.78 |
| | N2 | | | 300 | 392.78 |
| | | | | | |
| | N0 | | | 0 | 129.46 |
| W3 | N1 | 755.9 | 129.46 | 150 | 279.46 |
| | N2 | | | 300 | 429.46 |

---

## Author Comment (AC4) · 27 Nov 2016

This is my first time using this system and I want to be sure you can read well everything.

In the suplement file you will find the answer to the three referees, the new version and the certificate of the translator.

Thank you so much for your time.

Please also note the supplement to this comment:
http://www.nonlin-processes-geophys-discuss.net/npg-2016-32/npg-2016-32-AC4-supplement.zip